# Electric Vehicle Charging Load Prediction Model Considering Traffic Conditions and Temperature

**Jiangpeng Feng [1], Xiqiang Chang [1,2,*], Yanfang Fan [1] and Weixiang Luo [1]**

[1] College of Electrical Engineering, Xinjiang University, Urumqi 830047, China; xjufengjiangpeng@163.com (J.F.); 18733535871@163.com (Y.F.); lwx18839160071@163.com (W.L.)
[2] State Grid Xinjiang Electric Power Supply Company, Urumqi 830011, China
[*] Correspondence: 107552104061@stu.xju.edu.cn

**Abstract:** The paper presents a novel charging load prediction model for electric vehicles that takes into account traffic conditions and ambient temperature, which are often overlooked in conventional EV load prediction models. Additionally, the paper investigates the impact of disordered charging on distribution networks. Firstly, the paper creates a traffic road network topology and speed-flow model to accurately simulate the driving status of EVs on real road networks. Next, we calculate the electric vehicle power consumption per unit kilometer by considering the effects of temperature and vehicle speed on electricity consumption. Then, we combine the vehicle's main parameters to create a single electric vehicle charging model, use the Monte Carlo method to simulate electric vehicle travel behavior and charging, and obtain the spatial and temporal distribution of total charging load. Finally, the actual traffic road network and typical distribution network in northern China are used to analyze charging load forecast estimates for each typical functional area under real vehicle–road circumstances. The results show that the charging load demand in different areas has obvious spatial and temporal distribution characteristics and differences, and traffic conditions and temperature factors have a significant impact on electric vehicle charging load.

**Keywords:** electric vehicles; traffic conditions; Monte Carlo method; spatio-temporal distribution; load forecasting



## 1. Introduction

In recent years, with the increasingly stringent environmental requirements in countries, the electrification of transportation has been seen as an effective measure to achieve energy savings, emission reduction, and carbon reduction [1]. Electric vehicles are an important vehicle for promoting the transformation and upgrading of vehicle electrification in transportation systems [2]. By the end of 2022, the ownership of pure electric vehicles in China reached 10.45 million [3]. With the large-scale promotion and application in China, a large number of electric vehicles are linked to the power grid for charging, which not only raises the total demand for electricity but also intensifies the load peak-to-valley difference and has a more adverse effect on the safe and stable operation of the power distribution network [4]. Moreover, the charging demand of electric vehicles is also randomly influenced by random driving behavior characteristics, which makes the charging requirements of electric vehicles different from other loads, and the traditional load forecasting model is no longer applicable to the charging requirements of the electric vehicle [5]. The load forecasting model is not only the basis for assessing the ability of distribution networks to accept the electric vehicle but also a necessary prerequisite for promoting research on charging station planning and vehicle–road network integration and interaction, and it is also of great significance for the operation and construction of future power systems [6]. Based on the above reasons, it is necessary to establish a model that can precisely describe the spatio-temporal distribution characteristics of the electric vehicle charging demand.

On the basis of the survey data of the National Household Vehicle Travel Survey (NHTS) released by the United States [7], assuming that electric cars have the same travel pattern as fuel cars, the fitted curves of departure time, charging time, and daily driving distance are obtained. In [8,9], the Monte Carlo random sampling method is used to simulate the charging load distribution of electric vehicles. In [10,11], the spatial and temporal distribution characteristics of charging loads of different types of EVs in different charging areas and on different typical days are studied according to the scale of EVs and the development level of charging facilities in China and taking into account the randomness of EV movements. In addition, the impact of EV user behavior decisions and real-time charging tariffs on the spatial and temporal distribution of charging load is considered in [12]. In [13], a coupled system considering the "vehicle-charging station-distribution network" is constructed, and the spatio-temporal distribution of EV charging load is simulated by the source–end matrix, taking into account the coupling of EVs with the transportation network and distribution network. In [14], a prediction model of the spatial and temporal distribution of EV charging load is proposed based on the law of gravity, and the spatial and temporal distribution of EV charging load under the fusion of "vehicle-road-station-grid" multi-source information is simulated. Notwithstanding the above-mentioned research, which constructs a prediction model of EV charging load, it ignores the stochastic characteristics of the EV driving process and charging process and sets the characteristic parameters of EV charging location and charging duration as fixed values, which cannot truly reflect the specific driving behavior and charging process of EVs. To solve this problem, [15,16] proposes a spatial–temporal model of electric vehicles based on stochastic travel chains and Markov decision theory. In [17], the spatio-temporal distribution of EV charging demand is predicted by considering different dates and functional areas with the inclusion of random travel chains. Considering multiple charging behaviors and different destinations, a Monte Carlo approach is introduced to simulate the driving and charging behavior of electric vehicles in [18]. Despite the fact that the electric vehicle load forecasting model proposed has a certain effectiveness in the above study, the impact of traffic jams, ambient temperature, and other factors on the driving process of electric vehicles is not considered in predicting the charging load of electric vehicles, and the charging load of electric vehicles cannot be accurately predicted.

According to the analysis above, electric vehicles are closely coupled to the transportation system and the distribution network. For the sake of describing the travel and charging demand characteristics of the electric vehicle more realistically in the dimensions of time and space and making the emulation results more precise, the coupling relationship between traffic flow and electric current, as well as the influence of factors such as traffic jams and ambient temperature, on the driving process and charging process of the electric vehicle needs to be sufficiently studied. According to the above context, the paper proposes a spatio-temporal distribution prediction model for electric vehicle charging load considering road information and environmental temperature.

Compared with past studies, the contributions of this paper include the following:

(1) This study introduces the effects of traffic conditions and ambient temperature on EV electricity consumption into the field of EV charging load forecasting, which solves the problem of these two important factors of traffic conditions and ambient temperature being ignored in the modeling process of load forecasting and further improves the accuracy of electric vehicle charging load forecasting compared with previous modeling methods.

(2) Proposed the shortest travel time as the goal of the path planning algorithm, according to the current driving section of the road class and traffic density dynamic planning route, so that the user can effectively avoid the shortest distance but the road speed limit is low and the road conditions for the congested road, more in line with the actual vehicle driving law.

(3) The spatial–temporal distribution characteristics of electric vehicle charging load and its impact on the distribution network are analyzed to provide a basis for guiding

the orderly charging of electric vehicles and the rational planning of the distribution network in the future.

## 2. Electric Vehicle Driving Characteristics Modeling

Electric vehicles have the dual characteristics of traffic and mobile load and the driving characteristics of electric vehicles, and their charge–discharge behavior links the urban distribution network and transportation network together; the model of the coupled "vehicle-road-network" system is shown in Figure 1. The topology and traffic saturation of the urban road network directly affect the driving speed of electric vehicles and motorists' path decisions, which, in turn, affect the spatial and temporal distribution of charging loads and the state of the distribution network; therefore, it can be seen that the detailed portrayal of the trip characteristics of electric vehicles is the foundation for the establishment of the "vehicle-road-network" coupling system.

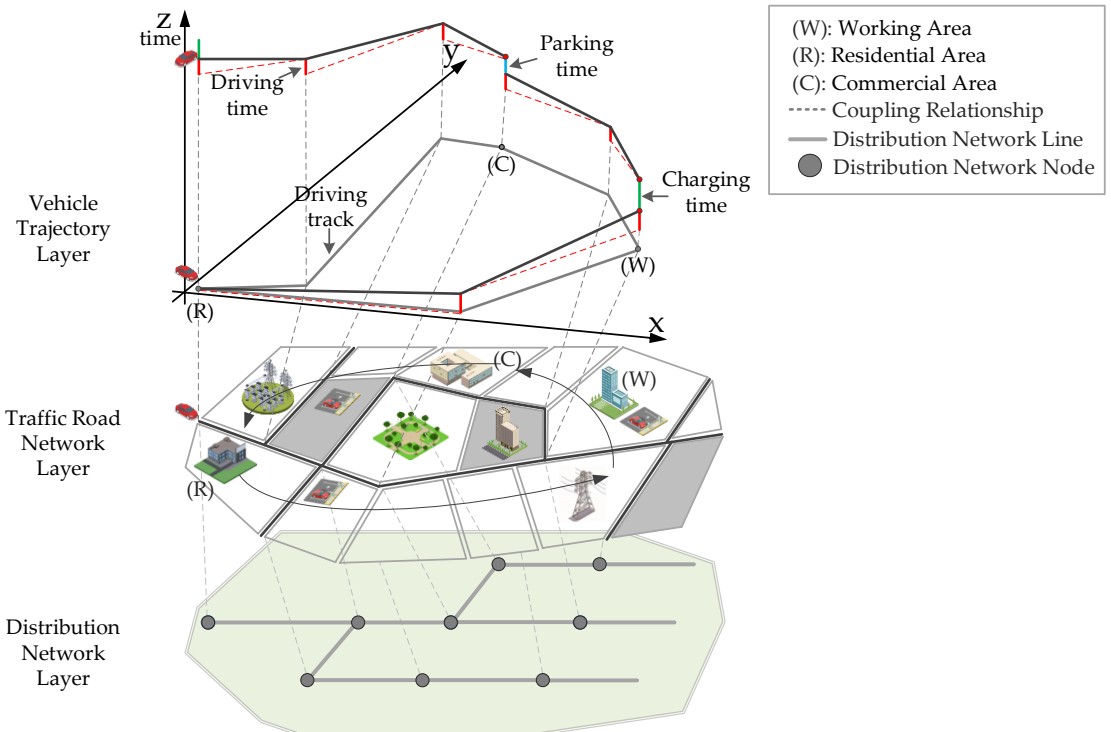

**Figure 1.** "Vehicle-road-network" coupling system.

### 2.1. Initial Travel Time

The initial travel time of private cars in a day is closely related to the spatio-temporal distribution of charging demand for the EV. Applying the statistical results of the 2017 NHTS, and using data fitting methods from study [19], the starting time of private car driving behavior in a day is extracted to study the travel characteristics of electric vehicles. The probability density function for the initial travel time of electric private cars can be calculated using real data. This probability distribution is then fitted, and it is discovered that the probability density function of the consumer's initial travel time is represented by Equation (1):

$$f(x_1) = \frac{1}{x_1 \sigma_1 \sqrt{2\pi}} \exp\left[-\frac{(x_1 - \mu_1)^2}{2\sigma_1^2}\right] \tag{1}$$

In Equation (1), $\mu_1$ is the expected value of the initial trip time, and $\sigma_1$ is the initial travel time's standard deviation fitted with $\mu_1$ = 7.89 and $\sigma_1$ = 1.96.

### 2.2. Driving Distance

The charging frequency and duration required for each charge for EV customers are closely related to driving distance, and there are obvious variances in driving distance across EV users due to various car habits. Furthermore, the driving distances of various types of electric vehicles vary greatly. Private automobile travel areas are primarily centered in cities, where cars are primarily used for commuting and amusement, and driving lengths are quite short and regular. As a result, the primary research object for this study is electric private cars. According to the 2017 NHTS statistics, the probability density function for the distance driven by electric private cars roughly fits a lognormal distribution, as shown in Equation (2):

$$f(x_2) = \frac{1}{x_2 \sigma_2 \sqrt{2\pi}} \exp\left[-\frac{(\ln x_2 - \mu_2)^2}{2\sigma_2^2}\right] \tag{2}$$

In Equation (2), $\mu_2$ is the driving distance's expected value, and $\sigma_2$ is the driving distance's standard deviation fitted with $\mu_2 = 3.68$ and $\sigma_2 = 0.88$.

### 2.3. Parking Time

The duration of EV consumers' stays at a destination location is highly tied to the function of the area in which the destination is located. When a user travels to a work area, the user's stay is longer due to the consumer's work needs; when the user travels to a commercial area for entertainment and consumption, the consumer's stay is shorter; and when the consumer travels to a residential area, the consumer's stay is also longer due to the consumer's home life needs. By fitting the statistical results of the 2017 NHTS, the dwell time $t$ of private cars in different areas is discovered to obey distinct types of generalized extreme value distributions. The generalized extreme value distribution is used to mathematically model the probability density of parking time in working areas, as represented in Equation (3):

$$\begin{cases} z_1 = \frac{t - 438.455}{164.506} \\ f(z_1) = \frac{1}{164.506} \exp\left[-(1 - 0.234 z_1)^{4.27}\right] \times (1 - 0.234 z_1)^{3.27} \end{cases} \tag{3}$$

The Weibull distribution is used to mathematically model the probability density of parking time in residential areas, as represented in Equation (4):

$$\begin{cases} z_2 = t \\ f(z_2) = \frac{1.153}{195.787} \exp\left[-\left(\frac{z_2}{195.787}\right)^{1.153}\right] \times \left(\frac{z_2}{195.787}\right)^{0.153} \end{cases} \tag{4}$$

The generalized extreme value distribution is used to mathematically model the probability density of parking time in commercial areas, as represented in Equation (5):

$$\begin{cases} z_3 = \frac{t - 68.520}{41.761} \\ f(z_3) = \frac{1}{41.761} \exp\left[-(1 + 0.657 z_3)^{-1.52}\right] \times (1 + 0.657 z_3)^{-2.52} \end{cases} \tag{5}$$

### 2.4. Initial State of Charge

At present, the range of electric vehicle batteries in China can meet the consumer's urban travel needs for more than two days, so the consumer's battery is not fully charged when he sets off initially, and the initial probability density function of SOC (state of charge) is as shown in Equation (6):

$$f(S_0) = \begin{cases} 4.352 S_0^{3.352}, 0 < S_0 \leq 1 \\ 0, S_0 = 0 \end{cases} \tag{6}$$

In Equation (6), $S_0$ is the initial battery SOC of the electric vehicle.

### 2.5. Travel Chain Model

Travel chains are used to describe the space movement process of electric vehicles within a day, including travel origins and destinations, because electric vehicles have spatial movement characteristics, and the spatial and temporal distribution of charging loads is affected by their uncertain movement process. Electric private automobiles travel to destinations with a high degree of regularity, and travel chains can be used to characterize this process. According to the functional differences of the destinations, they can be divided into three categories: the residential area (R), the working area (W), and the commercial area (C). These different types of areas also provide electric vehicle charging services. The three basic travel chain types discussed in this paper are R–C–R, R–W–R, and R–W–C–R, as shown in Figure 2.

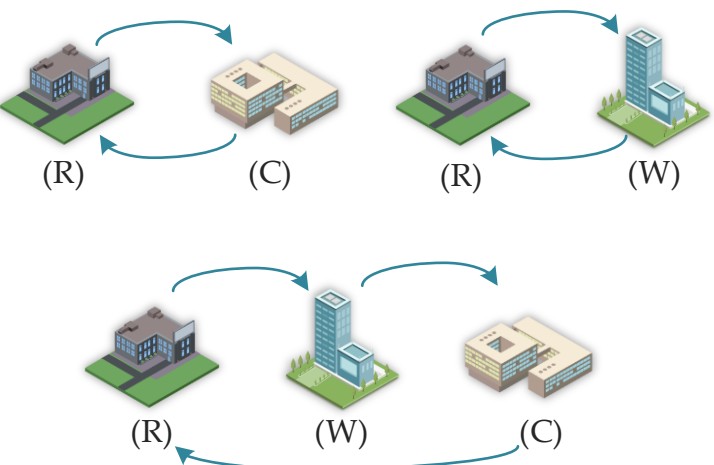

**Figure 2.** Different forms of travel chain structures.

The share of various travel chains for daily activities using private vehicles, as reported by the 2017 NHTS, is displayed in Table 1.

**Table 1.** Percentage of different forms of travel chain.

| Trip Chain Form | Specific Gravity/% |
|:---:|:---:|
| R − C − R | 23.1 |
| R − W − R | 52.8 |
| R − W − C − R | 24.1 |

## 3. Traffic Network and Distribution Network Modeling

### 3.1. Traffic Network Topology

In order to quantitatively explain the road structure in this paper, a graph theory approach is employed to model the two-way traffic network. The topology of the traffic network is shown by $E(G)$. $A(i,j)$ is the set of road weights of the traffic network in $E(G)$ used to describe the connection relationship between the nodes in $E(G)$; the relationship between $E(G)$ and $A(i,j)$ is as shown in Equation (7):

$$E(G) = [A(i,j)] \tag{7}$$

The specific values of the traffic network road weights $A(i,j)$ are as shown in Equation (8):

$$A(i,j) = \begin{cases} l_{ij}, (i,j) \in E(G) \\ 0, i = j \\ \inf, (i,j) \notin E(G) \end{cases} \tag{8}$$

In Equation (8), *i* and *j* both denote the nodes of $E(G)$ in the traffic road network; $l_{ij}$ denotes the distance between nodes *i* and *j* in $E(G)$; and inf denotes that the two road network nodes concerned are not adjacent to each other.

As an example, taking the traffic network depicted in Figure 3, the intersections of the six road sections constitute a traffic road network set whose corresponding matrix $E(G)$ is as shown in Equation (9):

$$E(G) = \begin{bmatrix} 0 & l_{12} & \text{inf} & l_{14} & \text{inf} \\ l_{21} & 0 & l_{23} & \text{inf} & l_{25} \\ \text{inf} & l_{32} & 0 & \text{inf} & l_{35} \\ l_{41} & \text{inf} & \text{inf} & 0 & l_{45} \\ \text{inf} & l_{52} & l_{53} & l_{54} & 0 \end{bmatrix} \tag{9}$$

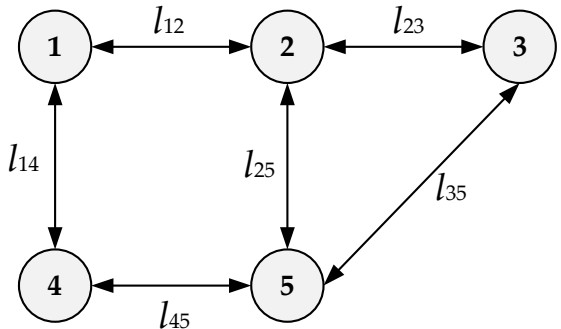

**Figure 3.** Road traffic network topology.

### 3.2. Improvement of Road Traffic Impedance Model

3.2.1. Improved Roadway Impedance Model

The travel time of electric vehicles on the road can be calculated according to the BPR (Bureau of Public Roads) model [20]. The road can be classified as smooth ($0 < S \leq 1.0$) or congested ($1.0 < S < 2.0$) by the road section saturation *S*. Since the road traffic network is dynamic, *S* varies over time. The equation of road saturation S is as shown in Equation (10):

$$S = \frac{Q}{C} \tag{10}$$

In Equation (10), *Q* is the section of road's traffic flow, which varies dynamically with time; and *C* is the section of road's maximum capacity.

According to the section of road's traffic condition, the road section impedance model corresponding to different road section saturation *S* is obtained. The function can reflect the change in road resistance with road congestion. Considering the dynamic flow and speed limit of the road, the improved equation is obtained [21], as shown in Equation (11):

$$T_r = \begin{cases} t_0(1 + \alpha(S)^\beta), 0 \leq S \leq 1.0 \\ t_0(1 + \alpha(2 - S)^\beta), 1.0 < S < 2.0 \end{cases} \tag{11}$$

In Equation (11), $\alpha$ and $\beta$ are the impedance impact factors, usually set to 0.15 and 4, respectively; $T_r$ is the actual time required to go through this section; $t_0$ is the zero-flow travel time required to go through this section, which can be obtained from Equation (12):

$$t_0 = \frac{L_{ij}}{V_0} \tag{12}$$

In Equation (12), $L_{ij}$ is the path length from road node *i* to node *j*; $V_0$ is the zero-flow velocity of the indicated road, and there are differences in the zero-flow velocity of different levels of roads.

### 3.2.2. Improved Node Impedance Model

Most current studies set the road parameters as static parameters, which makes it difficult to characterize the impact of road intersections on the whole process of vehicle traffic, so this paper uses a modified Webster model to calibrate the nodal impedance of road intersections [22], which can be expressed as shown in Equation (13):

$$T_c = 0.441 \left[ \frac{c(1-\lambda)^2}{2(1-\lambda S)} + \frac{S^2}{2q(1-S)} \right] + 6.996 \tag{13}$$

In Equation (13), $c$ is the signal period; $\lambda$ is the ratio of the effective green duration of a signal phase to the total duration of the cycle; $q$ is the vehicle arrival rate of the driveway; $S$ is the road saturation.

Considering that the time consumed in the process of vehicle travel is not only impacted by the road congestion situation but also by the control of road intersection signals, the road traffic impedance should include two parts, road section impedance and road node impedance, and then the improved road traffic model can be expressed as shown in Equation (14):

$$T_h = T_r + T_c \tag{14}$$

### 3.3. Path Selection Method

In graph theory, a common method for solving path planning problems is Dijkstra's algorithm. When the total length $h$ of the path is represented by $L_h$, the goal of the Dijkstra algorithm is to find the path that minimizes $L_h$ among all possible $h$ values, which ignores the impact of actual road traffic factors on the vehicle movement process [23]. In the actual driving process, users prefer to choose the path without congestion; if the Dijkstra algorithm's objective is the minimum value of $T_h$, where $T_h$ indicates the road traffic impedance, i.e., the total time consumed through path $h$, planning for the dynamic shortest journey time can be achieved, and electric vehicle users can combine road traffic information when they reach the road intersection, and, according to the current traffic congestion, the shortest travel time path can be replanned when the EV user reaches the road intersection.

### 3.4. Regional Distribution Network Model

On the basis of establishing a dynamic transportation network node model, the IEEE 33 standard node distribution network model is used to connect distribution network nodes and transportation network nodes. Figure 4 displays the topology diagram of the distribution network.

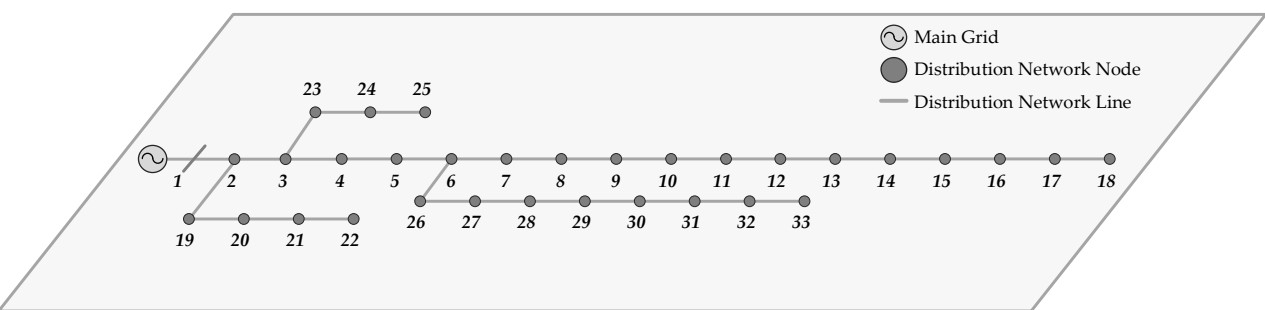

**Figure 4.** Power distribution network topologic model.

## 4. Electric Vehicle Charging Load Model

The spatio-temporal distribution of electric vehicle charging load in the transportation network is associated with the distribution of traffic travel. Electric vehicles are impacted by external environmental factors, including road speed limits, traffic congestion, road grade,

and ambient temperature during normal travel, which, in turn, affect the spatio-temporal distribution of electric vehicle charging time in terms of duration and charging location in the transportation network. By coupling the impact of external environmental factors on the electricity expenditure per unit kilometer to the process of electric vehicle travel, a more refined model of electric vehicle energy expenditure per unit kilometer considering traffic conditions and ambient temperature is established.

### 4.1. Model of Electric Vehicle Electricity Expenditure per Unit Kilometer Taking into Account Vehicle Speed and Temperature

4.1.1. Model Considering Road Conditions

Driving speed and battery energy expenditure of electric vehicles are linked with different road categories and under varying traffic jam scenarios. The electricity expenditure per unit kilometer for electric vehicles $E_i$ (i = 1, 2, 3, 4) can be calculated under real-time congestion conditions by calculating the measured data. $E_i$ (i = 1, 2, 3, 4) can be expressed by Equation (15) as follows:

$$\begin{cases} E_1 = \frac{1.52}{V} - 0.004V + 2.992 \times 10^{-5}V + 0.247 \\ E_2 = 0.004V + \frac{5.492}{V} - 0.179 \\ E_3 = -0.001V + \frac{1.531}{V} + 0.21 \\ E_4 = -0.002V + \frac{1.553}{V} + 0.208 \end{cases} \tag{15}$$

In Equation (15), $E_1$, $E_2$, $E_3$, and $E_4$ are the electricity expenditure per unit kilometer (kWh/km) of the vehicle traveling on the highway, main trunk, secondary main trunk, and branch way, respectively; $V$ indicates the real-time driving speed of the electric vehicle (km/h), which can be obtained from Equation (16):

$$V = \frac{L_{ij}}{T_h} \tag{16}$$

4.1.2. Model of On-Board Air Conditioner Considering Temperature Effects

As the largest single auxiliary load on electric vehicles, one of the influences of temperature on the on-board electric battery capacity is mainly embodied by the electricity expenditure of the air conditioner in summer and winter. When it is a high temperature inside the cabin in summer, owners may choose to open the on-board air conditioner to cool the vehicle, which increases the battery's electricity consumption. When temperatures inside the cabin are low in the winter, owners may choose to turn on the air conditioning to raise the temperature, resulting in increased battery electricity consumption. The above analysis shows that the operating state of the on-board air conditioner affects the battery electricity expenditure and thus the spatio-temporal distribution of the electric vehicle charging load.

The air conditioning electricity expenditure of the electric vehicle at distinct temperatures after the air conditioning is opened is as shown in Equation (17):

$$E_h = \begin{cases} 0.03347\frac{L}{V}(T_{\text{set}} - T_{\text{amb}})^{1.324}, T_{\text{amb}} < T_{\text{set}} \\ 0, T_{\text{amb}} = T_{\text{set}} \\ 0.03669\frac{L}{V}(T_{\text{amb}} - T_{\text{set}})^{1.084}, T_{\text{amb}} > T_{\text{set}} \end{cases} \tag{17}$$

In Equation (17), $E_h$ is the electricity expenditure of air conditioning when the ambient temperature is $T_{\text{amb}}$ and the vehicle travels $L$ km at speed $V$. The real-time vehicle speed $V$ can be obtained from Equation (16); $T_{\text{amb}}$ is the current ambient temperature; and $T_{\text{set}}$ is the vehicle cockpit set temperature, which is set to 22 degrees Celsius.

### 4.1.3. Model of Battery Losses in Electric Vehicles Considering Different Ambient Temperatures

In summer, when temperatures are high, the high external temperature causes the self-discharge of the vehicle battery to become more pronounced and the rate of battery aging to accelerate and causes problems such as restricted power and overheating protection, resulting in a certain degree of battery capacity loss. When the temperature is low in winter, the internal resistance of the battery increases, and the internal electrochemical process slows down, affecting the battery capacity and hindering charging and discharging performance. The above analysis shows that ambient temperature affects the battery performance of electric vehicles and therefore the spatio-temporal distribution of the charging load.

The loss per unit kilometer $E_t$ of the on-board battery of the electric vehicle without considering the air conditioning load as a function of the ambient temperature $T$ is as shown in Equation (18):

$$E_t = \begin{cases} -3.2547 \times 10^{-3}T - 6.3657 \times 10^{-5}T^2 + 6.5058 \times 10^{-6}T^3 + 2.4575 \times 10^{-7}T^4 \\ -8.4631 \times 10^{-9}T^5 - 1.9318 \times 10^{-10}T^6 + 0.21077, T_{\text{amb}} < T_{\text{set}} \\ 1.011 \times 10^{-2}T + 2.6278 \times 10^{-4}T^2 + 0.6777, T_{\text{amb}} \geq T_{\text{set}} \end{cases} \tag{18}$$

In Equation (18), $E_t$ is the power loss of the on-board battery due to temperature when the ambient temperature is $T_{\text{amb}}$ (not considering the air conditioning load).

### 4.1.4. Model of Electric Vehicle Electricity Expenditure Taking into Account Vehicle Speed and Temperature

From the above analysis, it can be concluded that vehicle speed and temperature jointly affect the electricity consumption per unit kilometer. Consequently, a model is established to count electricity consumption per unit kilometer considering vehicle speed and temperature, as shown in Equation (19):

$$E_T = E_h + E_t + E_i \quad (i = 1, 2, 3, 4) \tag{19}$$

In Equation (19), $E_T$ represents the real-time electricity consumption per unit kilometer of the electric vehicle when the vehicle speed is $V$ and the ambient temperature is $T$, unit: kWh/km.

### 4.2. Electric Vehicle Charging Method

The fast charging power is selected as 60 kW, and the slow charging power is selected as 7 kW. Considering the charging cost and battery loss of consumers, it is assumed that consumers prioritize the slow charging method. However, if the SOC does not reach the preset expectation during the parking time period, i.e., there is not enough power for the next leg of the electric vehicle's route, as shown in Equation (20), then the consumer will choose the quick charging method.

$$\frac{P_{\text{slow}} T_p^i}{C_{\text{EV}}} + soc_{\text{init}}^i < soc_{\text{exp}} \tag{20}$$

In Equation (19), $P_{\text{slow}}$ denotes the slow charging power; and $T_p^i$ denotes the parking time of the vehicle at node $i$.

Since the current quick charging technology can guarantee that the battery SOC of an electric vehicle with a capacity of 60 kWh can be charged from 0 to 80% in 20 min [24], the situation in which the SOC of the battery after quick charging is less than the SOC required for the next trip is ignored.

### 4.3. Electric Vehicle Charging Load Calculation

The spatio-temporal information of each distribution network node is counted based on the coupling relationship between transportation network nodes and distribution network nodes and then on statistics data on the charging load for the entire region.

According to the characteristic quantity of electric vehicles, the total charging load at node *i* of the distribution grid at time *t* is calculated. The entire load of the nodes in different functional areas is then calculated by adding the grid node's base load and the whole EV charging load, finally obtaining the spatio-temporal distribution of the EV charging load. The prediction model predicts the first 24 h at an interval of 1 min, as shown in Equation (21):

$$P_a^i = P_a^{ib} + \sum_{i=1}^{N} P_{j,t}^i (2 \leq i \leq 33, t, i \in Z) \tag{21}$$

In Equation (21), $P_a^i$ denotes the total load at the distribution network's node *i* of area *a*; $P_a^{ib}$ denotes the base load at the distribution network's node *i* of area *a*; *N* denotes the number of electric vehicles performing charging activities near the distribution network's node *i* at moment *t*; $P_{j,t}^i$ denotes the charging power of the electric vehicle with serial number *j* at the distribution network's node *i* at moment *t*.

### 4.4. Load Forecasting Model Solving Process

The Monte Carlo method is utilized to simulate the spatio-temporal distribution of charging load on the basis of the created model of movement characteristics and charge characteristics of electric vehicles. Figure 5 depicts the flow chart for the prediction solution of the spatio-temporal distribution of the charging load of electric vehicles.

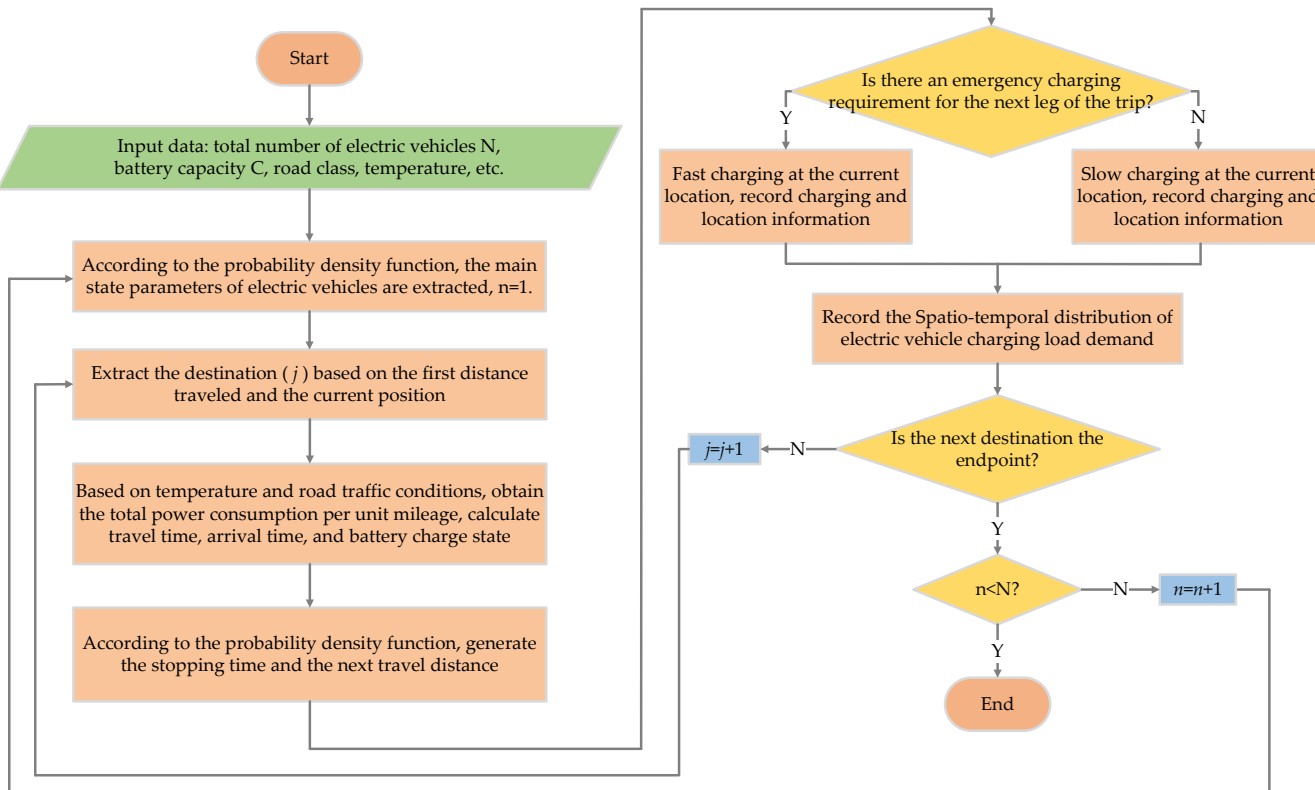

**Figure 5.** Flow chart of charging load spatio-temporal distribution prediction.

## 5. Example Analysis

To verify the validity of the proposed model, an actual urban road network in a region of Hebei Province in northern China is selected for simulation analysis.

*5.1. Simulation Parameter Setting*

The actual area indicated in Figure 6 is used as an example; the selected area's acreage is approximately 50.4 km², and the perimeter is approximately 30.6 km. The spatio-temporal distribution of electric vehicle charging load in various functional areas is calculated over a 24 h period.

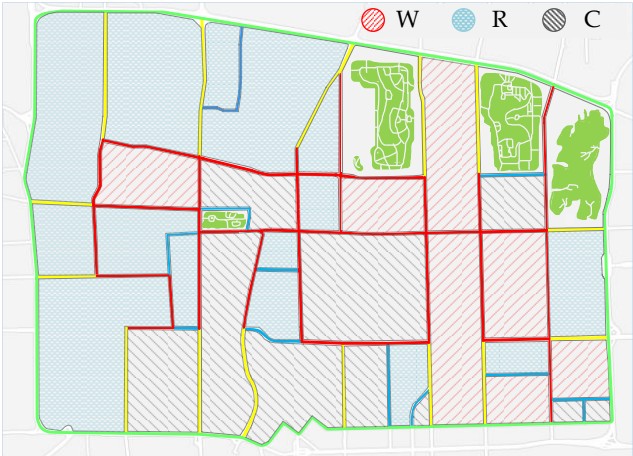

**Figure 6.** The streamlined transportation road network in the urban area.

(1)   Considering the main sections of the region, there are 64 traffic nodes and 98 traffic sections;

(2)   The actual area is simply divided into three functional categories, namely, residential, working, and commercial areas;

(3)   Combined with the practical road traffic situation in China, the roads are divided into four types, as shown in Figure 6: the ring highway that is the green road with a set zero-flow speed of 100 km/h; the main trunk that is the yellow road with a set zero-flow speed of 60 km/h; the secondary main trunk that is the red road with a set zero-flow speed of 50 km/h; the branch way that is the blue road with a set zero-flow speed of 40 km/h;

(4)   It is assumed that there are a total of 2500 electric private cars in this area whose initial locations are evenly distributed around the residential areas.

*5.2. Charging Load Forecast Results*

Figure 7 shows the total charging load of each functional area in the region. Subsequently, the typical nodes in each functional area are selected separately, and the characteristics of EV charging demand in different functional areas are analyzed by taking each typical node as an example.

Figure 8a shows what happens when electric vehicle charging load is linked to a typical node in a residential area. The residential area load increases comprehensively during all hours, showing an obvious double-peak phenomenon. Furthermore, the peak electric vehicle charging load is stacked on the peak residential basic load, causing the undesirable effect of adding a peak to a peak, resulting in a further increase in the load peak-to-valley difference, with the maximum peak-to-valley difference reaching 649 kW.

Figure 8b shows that the charging load in the working area begins to climb at 6:00 a.m. The charging load in the working area peaks after the morning peak commute to work, i.e., the peak load occurs at about 12:00 p.m.

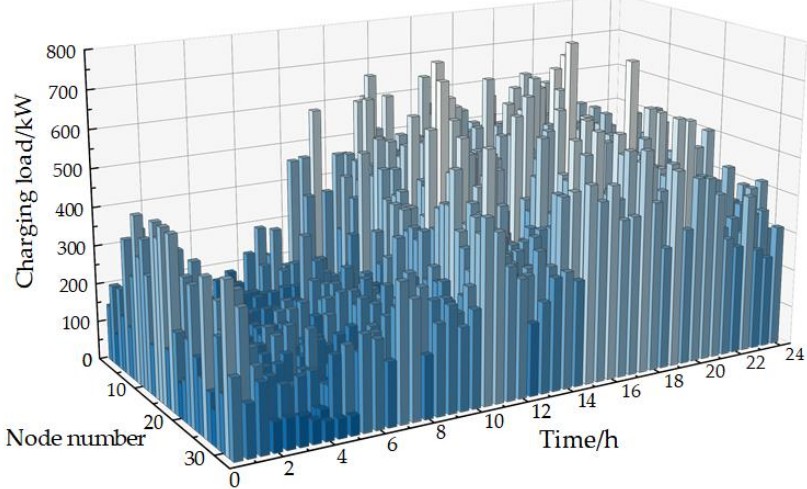

**Figure 7.** Spatial–temporal distribution of total charging load in various functional area.

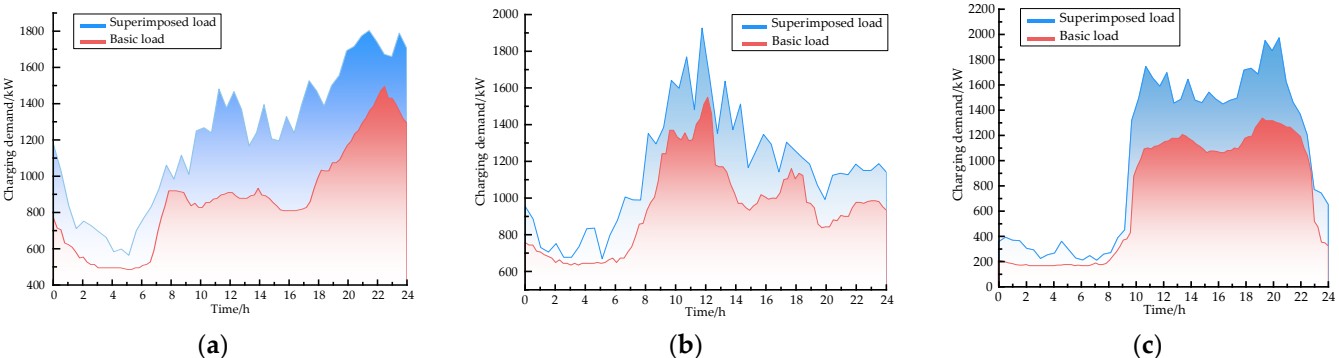

| (a) | (b) | (c) |

**Figure 8.** (**a**) Typical node load in the residential area; (**b**) typical node load in the working area; (**c**) typical node loads in the commercial area.

Figure 8c depicts the charging load distribution in the commercial area, with EV charging loads beginning to grow significantly at 12:00 p.m. and 20:00 p.m., the peak business hours in commercial regions, which is in line with the business characteristics of the commercial area.

A comprehensive comparison of Figure 8a–c shows that charging load is tightly coupled to travel destination and travel time. The charging load peaks in the working area after the morning rush hour and in the residential area after the evening rush hour. The peak charging load in the commercial area is concentrated during the midday meal and dinner times. The charging load in the residential, the commercial, and the working area increases to varying degrees after the EV charging load is linked, causing a significant change in the load curve, which has a significant effect on the regional distribution network.

Based on the emulation results, it can be seen that charging load prediction can provide an important reference for distribution network planning and safe and stable operation.

### 5.3. Path Selection Method

The path planning experiments designed in this paper do not take into account the costs incurred by electric vehicles in the process of finding charging stations and in the charging process; they only compare and verify the search effect and the path selection method presented in this paper and the traditional shortest path selection method. The path selection results are shown in Figure 9, and the comparison results are shown in Table 2.

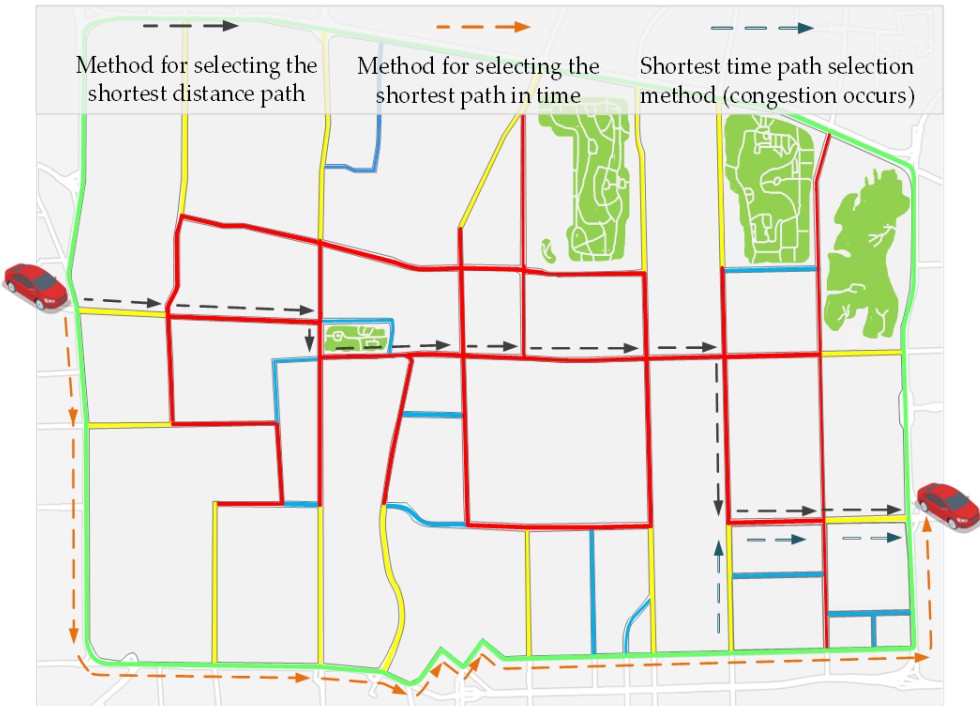

**Figure 9.** Comparison of path selection results.

**Table 2.** Comparison of different types of path selection.

| Path Selection Type | Path Length/km | Total Time/h |
|---|---|---|
| Shortest distance (open) | 35 | 0.764 |
| Shortest time (open) | 47.3 | 0.544 |
| Shortest time (congestion occurs) | 47.3 | 0.679 |
| Shortest time (change of path) | 46 | 0.646 |

As can be seen in Figure 9, the shortest time path selection method based on traffic information combines the road speed limit and real-time traffic situation in the path planning process to dynamically plan the driving path. When the road is clear, the shortest path selection method tends to select the road with a higher speed limit, as shown by the yellow arrow in Figure 9, and the selected road sections are all distributed on the ring road. The traditional shortest path selection method ignores the influence of road speed limits and chooses the road shown by the black arrow in Figure 9, and the selected road sections are mostly distributed on secondary arterial roads with lower speed limits, which are not given priority in the actual driving process for travel time. In addition, when congestion occurs in a certain road section, the shortest time path selection method presented in this paper modifies the path. For example, when congestion occurs in a certain section for the electric vehicle, the shortest path planning method modifies the original path shown by the yellow arrow to the new path shown by the blue arrow to avoid congestion. The different path passage times and path lengths are shown in Table 2.

Analysis of Table 2 shows that the path length planned by the shortest route selection method is 35.14% longer than that planned by the traditional shortest route selection method, but the shortest route selection method is 28.79% less than the traditional shortest route selection method in terms of travel time. Additionally, when a section of the planned route is congested, the replanned route consumes 5.11% less time than the original route which was previously congested, verifying that the shortest route selection method may make users effectively avoid paths with the shortest distance but low road speed limits and congested road conditions.

### 5.4. Consideration of the Impact of Traffic and Temperature Factors on the Charging Load

Taking into account the influence of traffic and temperature factors, the comparative results of EV charging load before and after the introduction of traffic and temperature in Figure 10 are obtained, and the proportion of energy consumption accounted for by different factors is obtained to compare and analyze the effect of traffic and temperature factors on charging load.

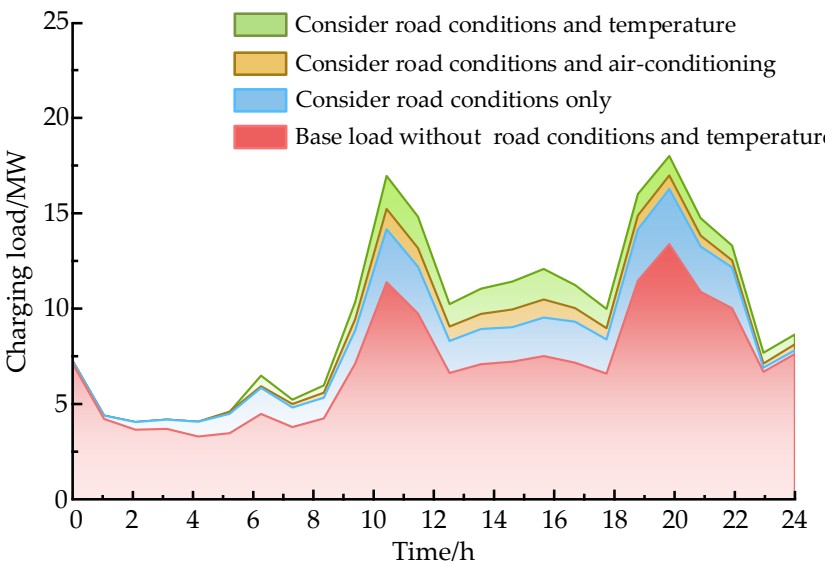

**Figure 10.** The effect of traffic and temperature factors on charging load.

From Figure 10, it can be seen that traffic conditions and ambient temperature factors have a greater impact on charging load. Taking the high-temperature environment in summer as an example, in the case of considering only the traffic road conditions, the peak hours of vehicle travel are 7:00–10:00 a.m. and 18:00–20:00 p.m., corresponding to the high saturation of the traffic road sections at this time. The traffic mileage consumption of vehicles in the process of driving increases, and the charging load starts to surge at this time. After 22:00 p.m., the saturation of traffic road sections is low, owing to the comparatively small number of vehicles on the road, and the energy consumption of traffic miles caused by traffic road conditions starts to decrease. In the case of considering only the traffic factor on the charging load, the charging load caused by the traffic road condition during the peak travel period increases by 30.63% compared with the case that does not consider the traffic road condition, obtaining an overall increase of 20.97%. In the case where only the ambient temperature causes the air conditioning to turn on, as the ambient temperature gradually increases over time, the electricity expenditure of the on-board air conditioning caused by the ambient temperature begins to rise continuously during the high-temperature hours of 10:00 a.m.–16:00 p.m., increasing the charging load by 12.90% compared to the charging load without the on-board air conditioning, with an overall increase of 6.52%. Considering only the battery loss due to ambient temperature, the charging load during the high-temperature period increases by 17.63% compared to that which does not consider the battery loss due to ambient temperature, obtaining an overall increase of 9.74%. The overall charging load increases by 38.28% compared to charging loads that do not take these two factors into account, verifying the validity of the model.

From Figure 11, the effect of different temperatures on the charging load can be seen; due to the difference in the cooling and heating principles of the on-board air conditioning of electric vehicles, the heating power of the on-board air conditioner in the winter low-temperature environment is higher than the cooling power of the on-board air conditioning in the summer high-temperature environment, and battery capacity loss due to temperature factors is higher in cold winter environments than in high-summer environments, resulting

in the winter peak charging load and the total charging demand in winter being higher than in summer, and the charging load caused by the winter low-temperature environment is 20.35% higher than that caused by the high summer temperature. The analysis shows that the low-temperature environment is particularly detrimental to the electric vehicle range, validating the model.

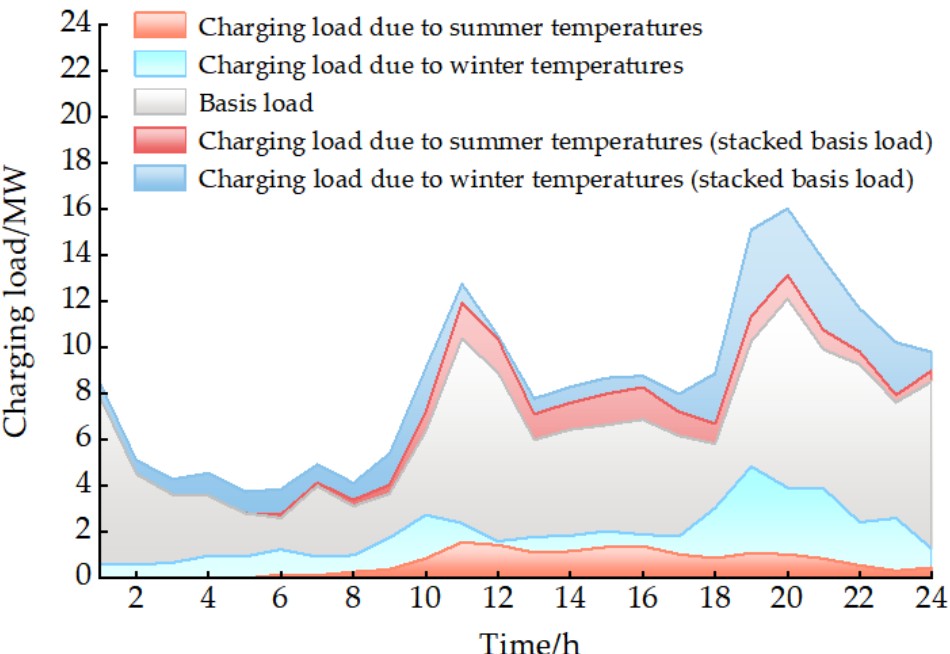

**Figure 11.** The effect of different temperatures on charging load are compared.

Comprehensive analysis of the simulation results shows that the dynamic energy consumption model established in this paper can accurately obtain the actual energy consumption of electric vehicles in different ambient temperatures and different traffic conditions, which further improves the accuracy of the electric vehicle load prediction results and is expected to promote the sharing of information and win–win cooperation between electric vehicle users, the traffic network, and the distribution network to improve the comprehensive benefits to society and provide a reasonable basis of planning for the future layout of the distribution network.

### 5.5. Analysis of the Impact of Uncontrolled Charging on the Distribution Network

In accordance with the spatial–temporal distribution of the EV charging loads obtained, an effort is made to assess the impact of macroscale electric vehicle disorderly charging on the distribution network.

From Figures 12 and 13, it is obvious that, with access to the EV charging load, the successive increase in load at the individual nodes in the different functional areas leads to a drop in the nodal voltage at each node.

As seen in Figure 14, node 18 is also the most affected as it is far away from node 1, the power node, and, as the electric vehicle is connected to node 18, the node voltage at this node drops from 0.9631 to 0.9447, which exceeds the lower limit of the node voltage.

A comprehensive analysis of Figures 12–14 shows that, during the peak electricity load period, with the connection of electric vehicle charging load, the node voltage of each node has a relatively large reduction which seriously affects the voltage quality for consumers.

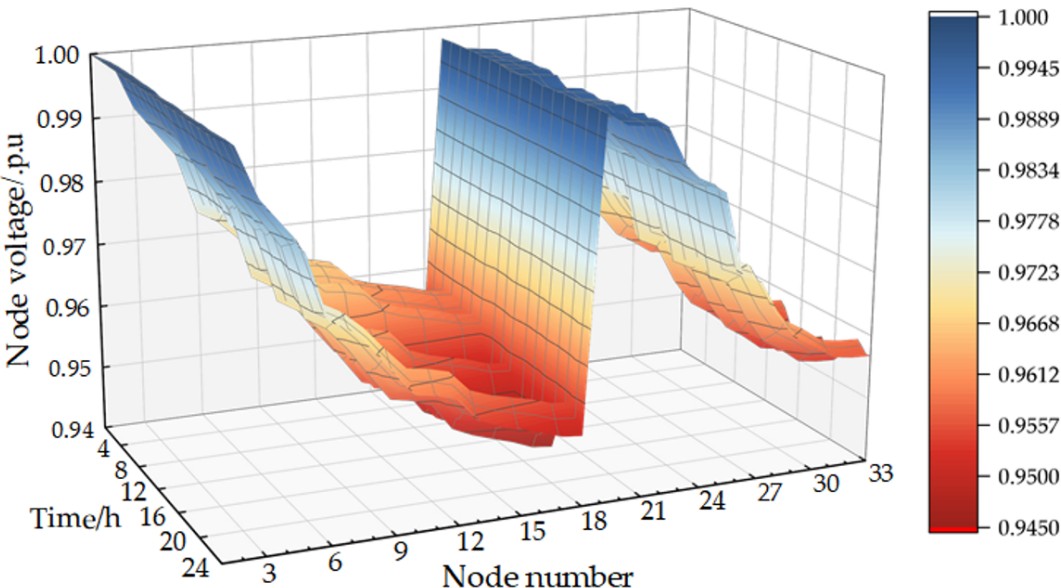

**Figure 12.** Nodal voltage at different moments after the charging load is linked to the distribution network.

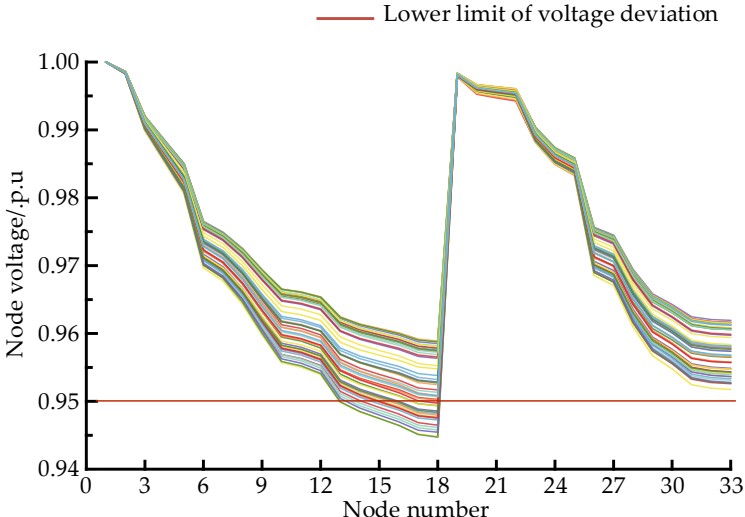

**Figure 13.** Voltage of each node after charging load is linked to the distribution network.

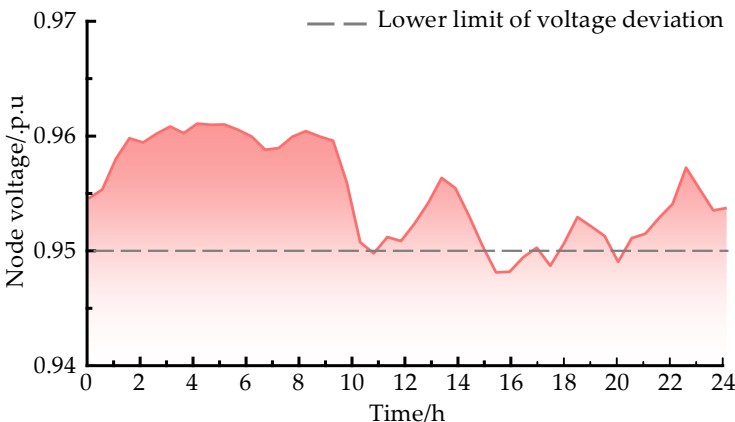

**Figure 14.** Node voltage at each moment of node 18.

Figure 15 clearly shows that, around 12:00 p.m. and 20:00 p.m., the overall network loss of the distribution network is larger because there are more EV charging loads at each functional area node during this time period. Under the base load, the peak losses are 0.199 MW and 0.172 MW, respectively, and when the charging load is linked, the peak losses are 0.220 MW and 0.193 MW, respectively, increasing by 10.6% and 12.5%. It can be seen that, along with the charging load being connected to the distribution network, the distribution network's network loss increases considerably, posing a serious threat to the distribution network's safe and economical functioning.

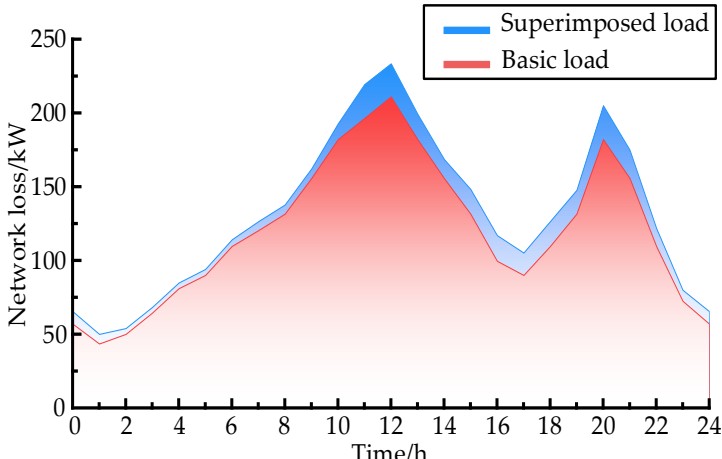

**Figure 15.** Distribution network loss.

## 6. Conclusions

In this paper, considering the effects of road congestion and ambient temperature on the electricity consumption of electric vehicles per unit kilometer, an electric vehicle charging load demand prediction model containing multiple stochastic processes is developed and solved using Monte Carlo simulation, which is used for predicting the spatio-temporal distribution of electric vehicle charging loads under the influence of road conditions and ambient temperatures, as well as investigating the impact of disordered charging on the distribution network. Through path planning experiments and spatio-temporal prediction of electric vehicle charging demand in different functional areas under actual road network distribution network conditions, the results verify the validity of the proposed model and method, and the conclusions and main contributions are as follows:

(1) The charging loads of different electric vehicles are obviously affected by the characteristics of functional areas, and there are obvious differences between the charging characteristics of different functional areas. The total charging load of electric vehicles has obvious "double-peak" characteristics, and, if not reasonably guided, it is superimposed on the peak value of the base load, resulting in an unfavorable impact;

(2) Electric vehicle users tend to choose roads with high traffic levels and flat road conditions for route planning. In order to simulate user behavior more realistically, this paper uses an improved shortest path algorithm with road weights as the objective function to plan the driving paths of electric vehicles, which reduces the traveling time by 28.79% compared with the traditional shortest path algorithm;

(3) Ambient temperature and traffic conditions have an impact on electric vehicle electricity consumption. In order to analyze the impact of these two factors on the charging load of electric vehicles, this paper introduces traffic conditions and ambient temperature in the modeling process, which can accurately obtain the actual electricity consumption of electric vehicles under different environmental conditions and improve the model prediction accuracy by 38.28% compared with the charging load prediction model which does not consider these two factors and provides a reasonable planning basis for the layout and capacity allocation of the distribution network.

Nevertheless, in the modeling process, this article assumes that there are charging stations at each node of the road network and that the capacity of the charging stations is large enough; therefore, the queuing time for charging a car and the process of the user selecting a charging station when charging is ignored, and further analysis is needed in the future taking into account the distribution of actual charging stations. In addition, due to the phenomenon that electric vehicle charging causes peak loads to superimpose on each other, a more effective, orderly charging strategy will be proposed in the future on the basis of the vehicle–road network coupling system to guide electric vehicles to charge in an orderly manner so as to achieve the effect of "peak load shifting".

**Author Contributions:** Conceptualization, J.F. and Y.F.; methodology, J.F.; software, J.F.; validation, J.F., Y.F. and X.C.; formal analysis, W.L.; writing—original draft preparation, J.F.; writing—review and editing, X.C.; visualization, W.L.; supervision, Y.F.; funding acquisition, X.C. All authors have read and agreed to the published version of the manuscript.

**Funding:** This research was funded by the Natural Science Foundation of Xinjiang Uygur Autonomous Region (2022D01C365), 2022 Tianshan Talent Cultivation Programme (2022TSYCLJ0019).

**Data Availability Statement:** The data presented in this study are available on request from the corresponding author. The data are not publicly available due to confidentiality agreements.

**Conflicts of Interest:** The authors declare no conflict of interest.

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
