# Peer review of "Electric Vehicle Charging Load Prediction Model Considering Traffic Conditions and Temperature"

_processes, doi:10.3390/pr11082256_

Round 1

Reviewer 1 Report

The paper "Electric Vehicle Charging Load Prediction Model Considering Traffic Conditions and Temperature" introduces a comprehensive and innovative approach to predict the charging load of electric vehicles (EVs). By incorporating traffic conditions and ambient temperature, the authors address two important factors that have traditionally been overlooked in conventional EV load prediction models. 

Point-to-point comments:

1.  Figure 1: The figure requires a more detailed description, including information about the coordinates, layers, and the meaning of the symbols R, W, and C. Providing additional explanations will help readers better understand the visual representation and its relevance to the paper's context.

2. Line 100, please cite the reference of the results.

3. Line 110: "Driving distance" would be a more suitable term to use instead of "distance traveled"

4. Line 117: The sentence "According to the 2017 NHTS statistics and then corrected with the actual situation in China" should be rephrased for clarity and grammatical correctness. Additionally, including the citation for the referenced results will enable readers to verify the accuracy of the information.

5. Line 128-129: The sentence appears to be redundant and could be revised or omitted to improve the flow and conciseness of the paragraph.

6. Line 131: Please ensure consistency in verb tense usage throughout the paper. Consider revising "was" to maintain a consistent tense with the rest of the text.

7. Line 132: There is a typo in the word "wherein." Please correct it for accuracy.

8. Line 132-134: The grammar in this sentence needs to be checked for clarity and correctness.

9. Line 139: It would be beneficial to provide an explanation of the acronym "SOC" to ensure readers unfamiliar with the term can fully understand the context.

10. Line 191: To support the parameter values mentioned, please provide appropriate references to the literature or sources from which these values were derived.

11. Line 228: Please provide a clear explanation of the information presented in Figure 4 to help readers understand its significance in relation to the paper's findings.

12. Line 268: To support the choice of the set temperature value, it would be helpful to provide references or justifications for the specific value used in the study.

13. Line 295: To support the selection of the maximum and minimum charging power values, please provide references or explanations.

14. Line 306: Please ensure consistent verb tense usage throughout the paper. Consider revising "was" to align with the overall tense used in the text.

15. Line 327: There are no figures 9 and 10. Please correct this error and ensure accurate referencing.

16. Highlighting the contributions of the paper and discussing how the proposed model can improve real-world scenarios would be beneficial. Additionally, conducting a more comprehensive analysis of the results and their implications will enhance the overall impact of the study.

Author Response

Dear reviewer, thank you for your detailed and professional advice, which helped us to improve the manuscript, we have made changes according to the reviewers' suggestions, the details of which are shown in the Word file, and we hope that we can get the reviewers' approval. Once again, I would like to express my appreciation for your efforts in refining this manuscript!

Reviewer 2 Report

This paper proposes a spatiotemporal distribution prediction model for electric vehicle charging load considering road information and environmental temperature. The article provides a detailed description of the travel characteristics of electric vehicles, followed by modeling the transportation network and distribution network, and calculating the load of electric vehicles in the constructed scenario. I’m more concerned about the following issues:

#1 In real life, the driving modes and routes of electric vehicles are diverse, and whether the vehicle road network coupling system is reasonable. It seems impossible to use a unified road model and power system model to handle all charging scenarios involving electric vehicles.

#2 In section 4.1, different calculation formulas were established in the electric vehicle power consumption per kilometer model considering vehicle speed and temperature. Please explain the rationality of the selected formula and parameter calculation.

#3 In section 4.1.2, the influence of temperature on the power consumption of on-board air conditioning batteries was considered, and the charging and discharging patterns of batteries vary in winter and summer. What I am puzzled about is why the curve trend of the calculated electric vehicle charging load in winter and summer is exactly the same.

#4 In section 5.5, the article analyzed the impact of disorderly charging on the power grid. We know that disorderly charging of electric vehicles represents users charging electric vehicles at any time, anywhere, and randomly. Please explain in detail how to consider disordered charging and how to calculate the disordered charging load in this article.

#5 It is recommended to match the warm and cold tones of the colors in Figure 13 with the seasons.

The English should be further improved to make it easier for readers to understand.

Author Response

(The authors gave the same response as above.)

Reviewer 3 Report

The paper provides an overview of novel charging load prediction model for electric vehicles. This paper should be of various researchers’ interest. The abstract is very good. Author does indicate the purpose of the article. The key words are well written. The quality of the language is appropriate, however, the author sometimes uses expressions in informal style and a few sentences are unclear to the reader. The review of the literature is fairly up-to-date. Unfortunately, the conclusion is so poorand clear. Autors should change a conclusion and will write a recommendation. The paper is of appropriate length and suggestions are well-written. 

The organization of sections is satisfactory  (Introducton,  research background, purpose of the article, Methodology/methods, outcome). 

In the theoretical section, the author do the business innovation model. References section is ok, but conclusion must be correct. The paper is of acceptable scientific quality, but the author must address the comments and suggestions. The paper requires small revision.

Author Response

(The authors gave the same response as above.)
